# Correlation of Somatostatin Receptor 1–5 Expression, [^68^Ga]Ga-DOTANOC, [^18^F]F-FDG PET/CT and Clinical Outcome in a Prospective Cohort of Pancreatic Neuroendocrine Neoplasms

**DOI:** 10.3390/cancers14010162

**Published:** 2021-12-29

**Authors:** Susanna Majala, Tiina Vesterinen, Hanna Seppänen, Harri Mustonen, Jari Sundström, Camilla Schalin-Jäntti, Risto Gullichsen, Jukka Schildt, Jukka Kemppainen, Johanna Arola, Saila Kauhanen

**Affiliations:** 1Department of Surgery, Division of Digestive Surgery and Urology, Turku University Hospital, University of Turku, P.O. Box 52, FIN-20521 Turku, Finland; risto.gullichsen@tyks.fi (R.G.); saila.kauhanen@utu.fi (S.K.); 2Turku PET Centre, Turku University Hospital, P.O. Box 52, FIN-20521 Turku, Finland; jukka.kemppainen@utu.fi; 3HUSLAB, HUS Diagnostic Center, Department of Pathology, Helsinki University Hospital, University of Helsinki, P.O. Box 400, FIN-00029 Helsinki, Finland; tiina.vesterinen@helsinki.fi (T.V.); johanna.arola@hus.fi (J.A.); 4Institute for Molecular Medicine Finland (FIMM), HiLIFE, University of Helsinki, P.O. Box 20, FIN-00014 Helsinki, Finland; 5Translational Cancer Medicine Research Program, Department of Surgery, Faculty of Medicine, Helsinki University Hospital, University of Helsinki, P.O. Box 340, FIN-00029 Helsinki, Finland; hanna.seppanen@hus.fi (H.S.); harri.mustonen@helsinki.fi (H.M.); 6Department of Pathology, Turku University Hospital, P.O. Box 52, FIN-20521 Turku, Finland; jari.sundstrom@tyks.fi; 7Abdominal Center, Division of Endocrinology, Helsinki University Hospital, University of Helsinki, P.O. Box 340, FIN-00029 Helsinki, Finland; camilla.schalin-jantti@hus.fi; 8Department of Clinical Physiology and Nuclear Medicine, Helsinki University Hospital, Haartmaninkatu 4, P.O. Box 340, FIN-00029 Helsinki, Finland; jukka.schildt@hus.fi; 9Department of Clinical Physiology and Nuclear Medicine, Turku University Hospital, P.O. Box 52, FIN-20521 Turku, Finland

**Keywords:** non-functional pancreatic neuroendocrine neoplasm, somatostatin receptors, positron emission tomography, immunohistochemistry, prospective study, surgical management

## Abstract

**Simple Summary:**

The need for prognostic and predictive biomarkers in pancreatic neuroendocrine neoplasms (PNENs) is great. Overexpression of somatostatin receptors (SSTRs) provides a molecular basis for imaging these tumors with ^68^Ga-labeled somatostatin (SST) PET/CT and for treatment with somatostatin analogs. We evaluated all 5 somatostatin receptors (SSTR1-5) with immunohistochemistry and prospectively compared the results with both [^68^Ga]Ga-DOTANOC and [^18^F]F-FDG PET/CT in a cohort of 21 non-functional (NF) PNENs. SSTR2 was the only SSTR subtype to correlate with [^68^Ga]Ga-DOTANOC PET/CT. High SSTR5 expression correlated with a low Ki-67 proliferation index, suggesting a better prognosis for these patients. Thus, our results confirm that SSTR2 has the highest impact on SSTR PET signaling of PNENs.

**Abstract:**

Purpose: The aim of this study was to correlate immunohistochemical (IHC) tissue levels of SSTR1-5 with the receptor density generated from [^68^Ga]Ga-DOTANOC uptake in a prospective series of NF-PNENs. Methods: Twenty-one patients with a total of thirty-five NF-PNEN-lesions and twenty-one histologically confirmed lymph node metastases (LN+) were included in this prospective study. Twenty patients were operated on, and one underwent endoscopic ultrasonography and core-needle biopsy. PET/CT with both [^68^Ga]Ga-DOTANOC and [^18^F]F-FDG was performed on all patients. All histological samples were re-classified and IHC-stained with monoclonal SSTR1-5 antibodies and Ki-67 and correlated with [^68^Ga]Ga-DOTANOC and [^18^F]F-FDG PET/CT. Results: Expression of SSTR1-5 was detected in 74%, 91%, 80%, 14%, and 77% of NF-PNENs. There was a concordance of SSTR2 IHC with positive/negative [^68^Ga]Ga-DOTANOC finding (Spearman’s rho 0.382, *p* = 0.043). All [^68^Ga]Ga-DOTANOC-avid tumors expressed SSTR2 or SSTR3 or SSTR5. Expression of SSTR5 was higher in tumors with a low Ki-67 proliferation index (PI) (−0.353, 95% CI −0.654–0.039, *p* = 0.038). The mean Ki-67 PI for SSTR5 positive tumors was 2.44 (SD 2.56, CI 1.0–3.0) and 6.38 (SD 7.25, CI 2.25–8.75) for negative tumors. Conclusion: SSTR2 was the only SSTR subtype to correlate with [^68^Ga]Ga-DOTANOC PET/CT. Our prospective study confirms SSTR2 to be of the highest impact for SST PET/CT signal.

## 1. Introduction

Gastroenteropancreatic neuroendocrine neoplasms (GEP-NENs) exhibit heterogeneous phenotypes with a variable clinical course ranging from very indolent tumors to highly aggressive carcinomas. Pancreatic neuroendocrine neoplasms (PNENs) have the lowest five-year survival rate (23%) of all GEP-NEN sites [1]. However, hormonally inactive, non-functional (NF) GEP-NEN may remain asymptomatic throughout life and have no clinical significance. GEP-NEN incidence has doubled from early 2000 to 2012 in the United States [2], which is partly due to the increasing use of imaging.

The 2017 update of the WHO classification incorporates a new subcategory of pancreatic neuroendocrine tumor grade 3 (PNET G3) into the well-differentiated NEN category [3]. Prognosis varies greatly depending on the proliferation activity of the tumor [4]. However, preoperative histopathological evaluation is often problematic, although non-invasive diagnostic imaging with [^18^F]F-FDG-PET/CT helps to predict the tumor grade better [5,6,7,8]. Our group previously showed that dual tracer imaging with [^68^Ga]Ga-DOTANOC and [^18^F]F-FDG can predict the proliferation index, thus the aggressiveness of NF-PNENs [8].

After decades of octreotide imaging, PET/CT has become the golden standard of imaging during the diagnostic work-up of PNEN. Performance of ^68^Ga-labeled somatostatin (SST) PET/CT is based on the membranous overexpression of somatostatin receptors (SSTRs) on the tumor cell. The physiological actions of somatostatin and its analogs (SSAs) are mediated through interactions with five G-protein coupled receptor subtypes: SSTR1, SSTR2, SSTR3, SSTR4, and SSTR5. Expression of SSTR subtypes varies greatly in human pancreatic islet cells [9].

Due to the clinical heterogeneity of NF-PNENs, it is crucial to find reliable markers that predict both prognosis and response to therapy. SSTR subtype analysis is a promising tool in the choice of treatment of NEN using novel SSAs. High SSRT2 expression has been suggested to be an even better predictor of overall survival in NENs than the Ki-67 proliferation index (PI) [10,11,12]. There are reports on the immunohistochemical (IHC) profile of SSTR1-5 in PNENs [13,14], but, to our knowledge, few published studies have analyzed all five human SSTRs of the tumor and their correlations with molecular imaging using PET/CT [15].

## 2. Materials and Methods

### 2.1. Patients

The SSTR analysis was performed as part of a prospective, controlled, two-center clinical trial at Turku and Helsinki University Hospitals in Finland. From 1/2016 to 1/2018, 35 patients suspected of having NF-PNEN after primary CT were prospectively imaged using [^68^Ga]Ga-DOTANOC-PET/CT and [^18^F]F-FDG-PET/CT. None of these patients had classical symptoms indicating a functioning PNEN, and all patients were evaluated by an endocrinologist at the corresponding university hospital. Patient selection is described in Figure 1 and patient characteristics in Table 1.

Twenty-one patients had a total of thirty-five histologically confirmed tumors, of which twenty-eight lesions were detectable upon PET/CT imaging. The median PET/CT imaging interval was 34 days (d) (IQR 9–76.5 d). In histopathological examination, six patients had stage I disease, seven had stage II disease (three IIA and four IIB), four had stage III disease, and two had stage IV disease (shown in Figure 1). Six patients had histologically verified lymph node metastases. The median primary tumor size of these patients was 49.5 mm (IQR 30.8–78.8 mm, range 24–90 mm), whereas the median tumor size of all patients was 20.0 mm (IQR 10–32.5 mm, range 5–100 mm). The follow-up time, mean 30.2 months (SD 6.2 m), was measured from the date of the first PET/CT scan to the review time. All tumors were assigned to two groups: a non-aggressive group and an aggressive group (shown in Table 2). Patients were treated in accordance with European Guidelines [16].

The study protocol was reviewed and approved by the Institutional Review Board of Turku University Hospital (ETMK 114/1801/2015). All patients provided written, informed consent before participating in the study. The study has been registered at ClinicalTrials.gov; Nonfunctional Pancreatic NET and PET imaging, NCT02621541.

### 2.2. [^68^Ga]Ga-DOTANOC and [^18^F]F-FDG-PET/CT Imaging

In Turku, PET/CT was performed by using the Discovery STE (*n* = 15) or VCT (*n* = 12) scanner (General Electric Medical Systems, Milwaukee, WI, USA) at the Turku PET Centre. PET/CT in Helsinki was performed by the Siemens Biograph mCT 64 (Siemens Healthineers, Erlangen, Germany) (*n* = 12) or the Gemini PET-CT scanner (Philips Inc., Columbus, OH, USA) (*n* = 1) at the Nuclear Medicine Department, Helsinki University Hospital, and by the Siemens Biograph 6 (*n* = 2) scanner (Siemens Medical Solutions, Malvern, PA, USA) at the Docrates Cancer Center, Helsinki. Patients underwent a whole-body PET/CT scan from the skull base to the mid-thigh, commencing 65 ± 15 min after the injection of [^68^Ga]Ga-DOTANOC and 54 ± 9 min after [^18^F]F-FDG. Analyses of PET/CT images were interpreted by a nuclear medicine physician (J.K), with short referral information but blinded to histopathological reports. The maximum standardized uptake values (SUVmax) were determined for every tumor or abnormal region for both [^68^Ga]Ga-DOTANOC and [^18^F]F-FDG PET/CT. PET/CT study areas with a focal activity greater than the background that could not be identified as physiological activity were considered to indicate tumor tissue. Lesions were graded on [^68^Ga]Ga-DOTANOC-PET/CT with the Krenning score by visually and semi-quantitatively comparing the SUVmax of the tumors to the liver and spleen reference organs [17]. NETPET score was defined by using the dual-tracer PET/CT [18]. The mean dose of intravenous [^68^Ga]Ga-DOTANOC was 142.7 ± 18.9 MBq, and [^18^F]F-FDG was 331.1 ± 58.5 MBq. The PET/CT protocol and data analysis have been described previously [8].

### 2.3. Immunohistochemistry

Briefly, fresh, 3.5 µm thick, whole slide tissue sections were deparaffinized, treated using heat-induced antigen retrieval, and then incubated with primary antibodies (shown in Table 3).

Immunoreactions were detected using either a polymer-based ultraView or OptiView Universal DAB Detection Kit (Ventana Medical Systems, Inc., Tucson, AZ, USA) or EnVision Detection System (Dako, Agilent Pathology Solutions, Santa Clara, CA, USA). Automated (BenchMark ULTRA, Ventana Medical Systems, Inc., Tucson, AZ, USA) or semi-automated (AutoStainer, Lab Vision Corp., Fremont, CA, USA) staining instruments were used. Appropriate positive controls (pancreas, small intestine) were used for each antibody.

### 2.4. Scoring of the Staining Results

Immunoreactivity for SSTR2 was classified solely by membranous staining with a scoring system introduced by Elston et al. [19] and Körner et al. [20]. Briefly, samples were scored as: negative (0) for no membranous staining, weak (1) for partial membranous positivity for <10% of the tumor cells, and moderate (2) partial membranous positivity for ≥10% of the tumor cells. A strong (3) score was assigned when circumferential membranous positivity was observed on tumor cells, and an intense (4) score when >95% of the tumor cells presented a strong, circumferential staining pattern. Additionally, cytoplasmic immunoreactivity in SSTR1 and SSTR3-5 staining were evaluated using the following scoring: negative (0), weak intensity (1), moderate intensity (2), and strong intensity (3). The scoring of the overall expression is presented in Figure 2.

The proliferation index based on the Ki-67 immunoreactivity in the nuclei was determined using ImmunoRatio image analysis software to evaluate the highest labeling region of at least 2000 cells [21]. Scoring was performed by two investigators (J.A. and T.V.) without knowledge of the clinical parameters. Representative images of immunohistochemical labeling of SSTR1-5 and Ki-67 are presented in Figure 3a–f.

### 2.5. Statistical Analysis

Normally distributed variables were expressed as means and standard deviations (SD), variables not following normal distribution as medians and interquartile ranges (IQR), and categorical variables as frequencies and proportions. The Kolmogorov–Smirnov test was used to test deviations from the normal distribution. The Spearman’s rank correlation was used to test the relationship between Ki-67 PI and SSTR expression as normally distributed data were lacking. The Mann–Whitney U or the Kruskal–Wallis tests were used to evaluate differences between the groups in continuous variables. The Fisher’s exact test was used for binary variables, the linear-by-linear association test for ordinal variables, and the Jonckheere–Terpstra test between continuous and ordinal variables. The level of statistical significance was set at 0.05. Two-tailed tests were used. Data analyses were performed by a statistical expert (H.M.) using IBM SPSS Statistics for Windows, Version 24.0 (IBM Corp., Armonk, NY, USA).

## 3. Results

### 3.1. SSTR2 Expression Correlates with [^68^Ga]Ga-DOTANOC PET/CT

Immunohistochemical expression of SSTR2 was the highest in NF-PNENs (91%) (Figure 4).

There was a correlation between positive/negative [^68^Ga]Ga-DOTANOC and immunohistochemical SSTR2 expression (Spearman’s rho −0.382, *p* = 0.043) (shown in the Appendix A).

When SSTR2 immunohistochemical expression was scored either positive or negative, all [^68^Ga]Ga-DOTANOC-avid tumors (*n* = 22/23) expressed membranous SSTR2 and the only [^68^Ga]Ga-DOTANOC-negative tumor did not express membranous SSTR2 (Spearman’s rho −1.000, *p* = 0.043). However, there was no association between SSTR2 expression and uptake intensity of [^68^Ga]Ga-DOTANOC, expressed as SUVmax (Appendix A). SSTR2 expression (positive/negative) correlated with the NETPET score. Of the 14 tumors that expressed SSTR2, 1 had a NETPET score of 1, 6 had a NETPET score of 2, 1 had a NETPET score of 3, 1 had a NETPET score of 4, and the only tumor classified NETPET score of 5 did not express SSTR2 (Spearman’s rho −0.406, *p* = 0.043). We also found a correlation between SSTR2 (positive/negative) and the Krenning score. Of the tumors that expressed SSTR2, 9 had a Krenning score of 4, 13 tumors had a Krenning score of 3, and only the tumor with a Krenning score of 2 did not express SSTR2 (Spearman’s rho 0.405, *p* = 0.043).

### 3.2. SSTR1, 3, 4, 5 Expression Does Not Correlate with [^68^Ga]Ga-DOTANOC PET/CT

Overall, expression of SSTR1, SSTR3, SSTR4, and SSTR5 were detected in 74%, 80%, 14%, and 77% of NF-PNENs, respectively. Details of the membranous, cytoplasmic, and overall expression of SSTR1-5, as described in the methods section, are shown in Figure 4.

There was no correlation between [^68^Ga]Ga-DOTANOC PET/CT and SSTR1 or SSTR3-5 immunohistochemical expression. There was no correlation between SSTR1 or SSTR3-5 expression profile and NETPET score or between SSTR1 or SSTR3-5 expression profile and Krenning score. Immunohistochemical SSTR1-5 membrane expression profiles versus [^68^Ga]Ga-DOTANOC uptake of all tumors are presented in Figure 5a–e.

### 3.3. [^68^Ga]Ga-DOTANOC-Avid Tumors

All 23 [^68^Ga]Ga-DOTANOC-avid tumors expressed SSTR2, SSTR3 or SSTR5, in any combination: 12 tumors expressed all 3 receptors, 7 expressed 2 of these receptors, and 4 expressed 1 receptor. However, the [^68^Ga]Ga-DOTANOC-negative tumor expressed only SSTR5. Eventually, it metastasized to the lymph nodes.

### 3.4. SSTR1-5 Expression and Correlation to [^18^F]F-FDG PET/CT

Thirteen tumors (59%) were [^18^F]F-FDG-negative. There was a negative correlation between the uptake of [^18^F]F-FDG and SSTR5 membranous expression (score 0–4); SSTR5 score 0 median SUVmax 5.2, range 3.4–12.6 (*n* = 3), score 1 median SUVmax 2.8, IQR 1.5–6.5 (*n* = 5), score 2 SUVmax 8.7 (*n* = 1), score 3 median SUVmax 2.6, IQR 1.7–3.1 (*n* = 13), score 4 SUVmax 1.4 (*n* = 1) (*p* = 0.033). No correlation between [^18^F]F-FDG PET/CT imaging and immunohistochemical SSTR1-4 expression was apparent.

### 3.5. Somatostatin Receptors and Proliferation Index

There was a negative correlation between SSTR5 expression and Ki-67 PI (membranous and overall expression, Spearman’s rho −0.360, CI −0.670–0.033, *p* = 0.034/−0.353, CI −0.654–0.039, *p* = 0.038). The mean Ki-67 PI for SSTR5 positive tumors was 2.44 (SD 2.56, CI 1.0–3.0) and 6.38 (SD 7.25, CI 2.25–8.75) for negative tumors. This correlation persisted when tumors were divided into WHO grades according to Ki-67 PI (−0.419, *p* = 0.005/−0.391, *p* = 0.004) (Figure 6).

Grade 1 tumors were positive for SSTR5 in 91% (*n* = 20) (Figure 7a–h) and grade 2–3 tumors in 54% (*n* = 8). Interestingly, there was no association between SSTR1, SSTR2, SSTR3, or SSTR4 expression and Ki-67 PI or tumor grade (*p* = 0.004).

### 3.6. SSTR Expression and Tumor Aggressiveness

Tumors were divided into two groups according to aggressiveness (Table 2), the non-aggressive group (*n* = 19) had higher SSTR5 and SSTR1 expression than the aggressive (*n* = 16) tumor group. Of the non-aggressive tumors, 95% (*n* = 18) expressed SSTR5 (*p* = 0.013) and 89% (*n* = 17) expressed SSTR1 (*p* = 0.0498) (Figure 8). Moreover, 84% (*n* = 16) of non-aggressive and 31% (*n* = 5) of the aggressive tumors expressed both SSTR5 and SSTR1 (*p* = 0.002) (Figure 8). There was no association between other SSTR subtypes and tumor aggressiveness.

Six patients had a total of twenty-nine lymph node metastases (mean 4.8, SD 3.4 nodes). When comparing the SSTR expression level of primary tumor and metastasis, no difference was found, except for cytoplasmic SSTR3 expression. It was higher in metastases (median increase by two scores, IQR 1.5–2.0, *p* = 0.031, paired Wilcoxon rank test, results within patients were averaged) (shown in Table 4).

## 4. Discussion

This prospective study included 35 surgically resected NF-PNEN lesions in 21 patients. IHC levels of SSTR1-5 subtypes were compared with [^68^Ga]Ga-DOTANOC and [^18^F]F-FDG PET/CT results. There was a positive correlation between [^68^Ga]Ga-DOTANOC PET/CT and SSTR2 expression, and a negative correlation between the uptake of [^18^F]F-FDG and SSTR5 expression.

Previous studies on correlations between SSTR1-5 expression and modern SSTR PET/CT in GEP-NENs are scarce, and all are retrospective [15,22,23,24,25,26]. Kaemmerer et al. [15] studied the expression of all five SSTRs using a novel monoclonal antibody for analyzing SSTR2a and polyclonal antibodies for SSTR1 and SSTR3-5 in 17 NENs and compared results with [^68^Ga]Ga-DOTANOC PET/CT. They found that both SSTR2a and SSTR5 expression correlated positively with the SUVmax of [^68^Ga]Ga-DOTANOC PET/CT. They excluded lesions <15 mm because of the partial volume effect. Another study of 19 PNEN patients found both the SUVmax and SUVmean of SSTR PET/CT correlated positively with SSTR2 mRNA in real-time polymerase chain reaction and SSTR2 protein by IHC [22]. Miederer et al. compared the scores of SSTR2 IHC (0–3) of 18 heterogenic NEN patients with SUV values of [^68^Ga]Ga-DOTATOC PET/CT and found a correlation [23]. Olsen et al. found gene expression of SSTR2 correlated positively with [^68^Ga]Ga-DOTANOC uptake among neuroendocrine carcinoma patients (*n* = 21) [24]. Gene expression leading to protein synthesis is, nevertheless, controversial. In a small study of 14 GEP-NENs, immunoreactive scores of SSTR2 and SSTR5 correlated with the SUVmax of [^68^Ga]Ga-DOTANOC PET/CT [25]. The correlation between SSTR2 expression and [^68^Ga]Ga-DOTANOC PET/CT is supported by our findings.

However, we found no correlation between the uptake of [^68^Ga]Ga-DOTANOC and SSTR1-5. To our knowledge, this is the first study to correlate SSTR PET/CT with all SSTR1-5 IHC in PNEN patients. All the [^68^Ga]Ga-DOTANOC-avid tumors expressed SSTR2, and the only [^68^Ga]Ga-DOTANOC negative tumor did not express SSTR2. Nonetheless, no statistically significant correlation between SSTR immunoactivity scores and [^68^Ga]Ga-DOTANOC uptake (SUVmax) was detected. This may be due to our small cohort size and to the more heterogenic patient cohorts of the former studies. Further, some [23,25] of the former studies used polyclonal antibodies, others [15,26] a combination of polyclonal and monoclonal antibodies, and one [24] evaluated gene expression instead of IHC, whereas we used only monoclonal antibodies. Perhaps, the wider binding affinity of [^68^Ga]Ga-DOTANOC to different SSTR subtypes compared to other [^68^Ga]Ga-SSTR-ligands affected our results, too.

Previous studies revealed SSTR2 to be a surrogate of better prognosis for NEN patients [10,11,14,15,25]. However, we found no association between SSTR2 and proliferation or other markers of aggressive tumors. Our findings concord with those of Kaemmerer et al. [22]. The result might be due to a high number of SSTR2 positive tumors (91%) and to a relatively high percentage of MEN1 patients having a favorable prognosis to start with. On the other hand, we found a correlation of SSTR5 expression with low proliferation and negative [^18^F]F-FDG PET/CT imaging; both results suggest a better prognosis. We previously showed [8] that negative [^18^F]F-FDG PET/CT predicts better outcomes for the PNEN patients. Here, we show that SSTR5 expression also correlates with a better prognosis. It is known that SSTR2 and SSTR5 downregulate pancreatic carcinogenesis and angiogenesis and initiate apoptosis [27,28,29]. SSTR5 is a predominant inhibitory receptor for glucose-induced insulin secretion and conveys most of the negative impacts of SSAs on glucose metabolism, but how this affects tumor proliferation is unknown [30].

Some study groups [15,22,24] have suggested that for patients with missing preoperative SSTR PET/CT, the use of SSTR2 IHC could be valuable and, thus, SST PET/CT could be useful in restaging and follow-up. Moreover, SSTR expression could affect theranostics and therapy by indicating the use of peptide receptor radionuclide therapy and SSA treatment. Interestingly, pasireotide has a 39-fold affinity to SSTR5 compared to octreotide or lanreotide [31,32]. Metastasis samples in our cohort had increased SSTR3 cytoplasmic expression compared with primary tumor samples. SSTR3 expression indicated an increased risk for shorter survival [33] and correlated with lymph node metastases [34] among pulmonary carcinoid-tumor patients. PNENs that have a worse prognosis than other GEP-NENs show a higher SSTR3 expression than other NENs [15,35]. Our results suggest that SSTR3 expression might be a significant factor in PNENs transformation into a more aggressive form. However, this interpretation must be validated in a larger cohort of PNEN patients.

The limitation of this present study of rare cancer is the small number of patients; however, the total number of tumors analyzed was 56, including 35 primary tumors and 21 metastases. The cohort size impairs the sensitivity and specificity of SSTR-profile to detect potentially aggressive tumors. We also had a low number of patients with metastasized PNENs and G3 PNENs. Another limitation is the relatively short follow-up time (mean 30.2 m, SD 6.2 m), which was insufficient to estimate a definitive prognosis. However, this is a consequence of the prospective nature of this study.

The strengths of the present study are the prospective design, along with the combination of molecular imaging with two tracers, [^68^Ga]Ga-DOTANOC and [^18^F]F-FDG. Another strength is the detailed morphological analysis using novel monoclonal antibodies and optimized IHC staining protocols [36]. In general, methods for IHC labeling of SSTRs and scoring criteria vary greatly. Staining results are also highly dependent on the quality of the primary antibody used. Monoclonal UMB antibodies have shown mostly membranous staining [36,37], an observation supported in our cohort, too. SSTRs are membranous receptor proteins, but cytoplasmic expression can be seen as a consequence of the internalization of receptors after binding to a natural ligand, e.g., SSA [38]. This is the rationale behind the analyses of both membranous and cytoplasmic expression.

## 5. Conclusions

In conclusion, our study offers data on immunohistochemical SSTR1-5 expression patterns in NF-PNENs in a prospective setting and confirms that SSTR2 correlates with positive/negative [^68^Ga]Ga-DOTANOC PET/CT. SSTR5 expression is associated with low Ki-67 PI and might thus associate with a better prognosis. Further prospective studies, in larger tumor series, are needed to study the correlation of SSTR expression profile with ^68^Ga-labeled SST and [^18^F]F-FDG PET/CT for the personalized management of PNEN patients.

## Figures and Tables

**Figure 1 cancers-14-00162-f001:**
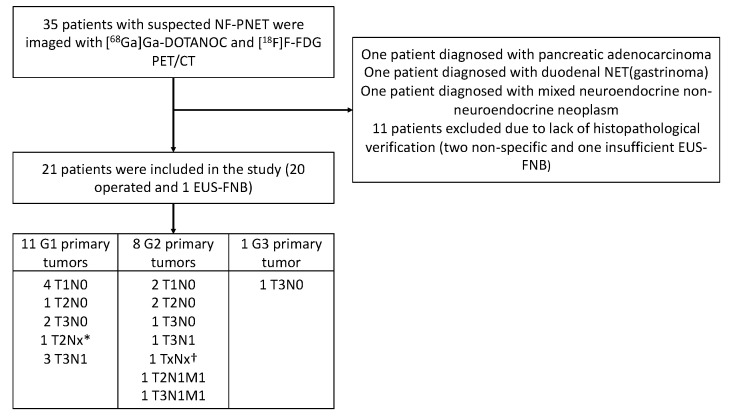
Flowchart of the recruitment and imaging of the patients and final histopathological findings (grade and stage) of the tumors. EUS-FNB, endoscopic ultrasonography and fine-needle biopsy. * Unknown lymph node status due to enucleation. † Diagnosis was made by biopsy, stage unknown.

**Figure 2 cancers-14-00162-f002:**
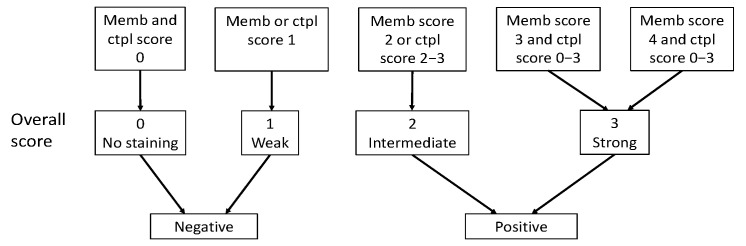
Classification of SSTR overall score as positive or negative based on membranous score and cytoplasmic score. Abbreviations: Memb, membranous; Ctpl, cytoplasmic.

**Figure 3 cancers-14-00162-f003:**
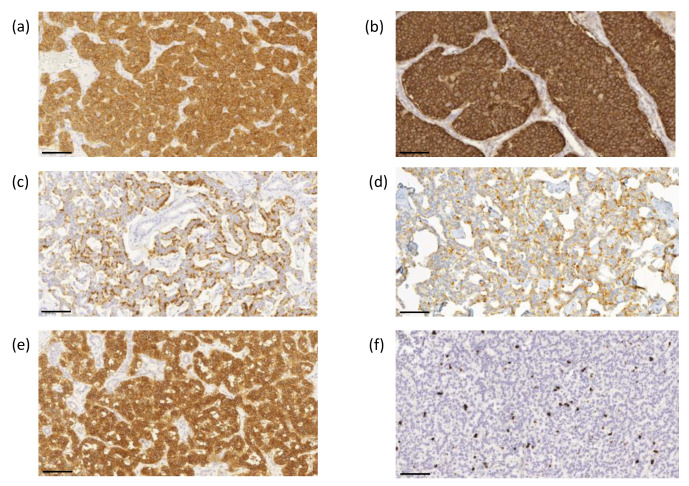
Representative images of immunohistochemical labeling of SSTR1-5 and Ki-67. SSTR1 (**a**) shows both membranous and cytoplasmic expression, whereas SSTR2 (**b**) and SSTR5 (**e**) expressions were mainly membranous. SSTR3 (**c**) and SSTR4 (**d**) show both membranous and cytoplasmic expression, often luminal. Ki–67 (**f**) expression was located in the nucleus. Images were obtained by the CaseViewer software 2.4 (3D HISTECH, Budapest, Hungary) with a magnification of 20×. Scale bar 100 µm.

**Figure 4 cancers-14-00162-f004:**
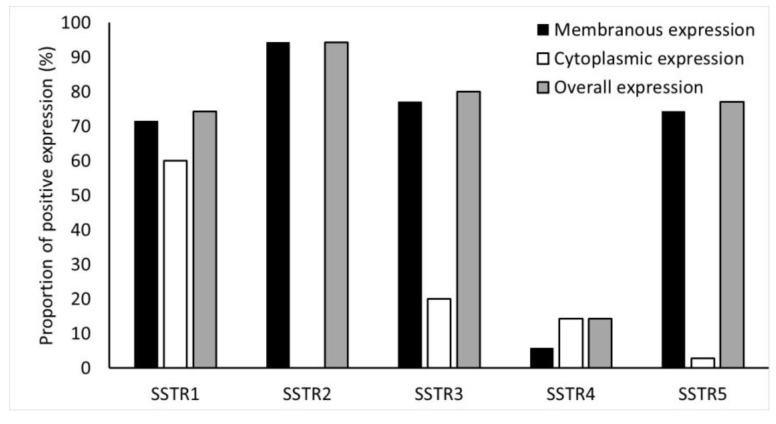
Frequency (%) of SSTR1-5 expressions in the tumors. SSTR2 staining was completely membranous.

**Figure 5 cancers-14-00162-f005:**
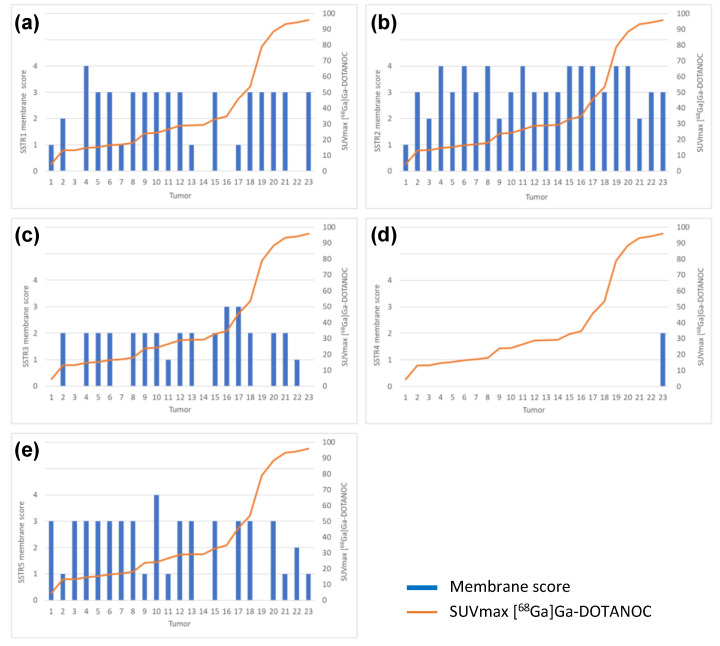
Immunohistochemical (**a**) SSTR1, (**b**) SSTR2, (**c**) SSTR3, (**d**) SSTR4 and (**e**) SSTR5 membrane expression profiles versus [^68^Ga]Ga-DOTANOC uptake by all tumors analyzed (*n* = 23).

**Figure 6 cancers-14-00162-f006:**
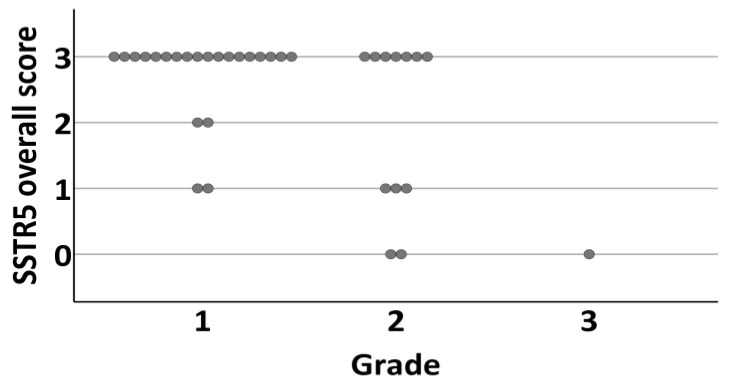
Distribution of tumors overall SSTR5 expression (0 = negative, 1 = weak expression, 2 = moderate expression, 3 = strong expression) according to grade (1–3) of the tumor.

**Figure 7 cancers-14-00162-f007:**
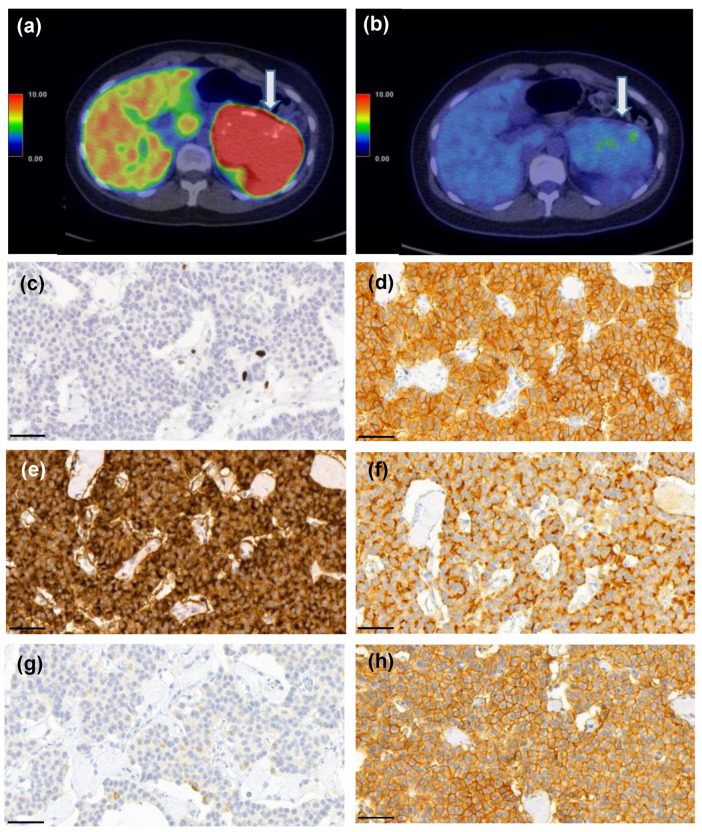
A 25-year-old female patient who had MEN1 syndrome had a 10-cm mass in the tail of the pancreas. [^68^Ga]Ga-DOTANOC-PET/CT (**a**) showed intense uptake (SUVmax 93.3 g/mL) and ^18^F-FDG-PET/CT (**b**) was also positive (SUVmax 4.5 g/mL vs. liver background activity SUVmax 3.6 g/mL). She underwent a distal pancreatectomy and splenectomy and histopathological analysis revealed a G1 T3N0 PNEN (Ki-67 PI 1%, **c**). Strong expression of somatostatin receptors (SSTR) 1 (**d**), 2 (**e**), 3 (**f**), and 5 (**h**) were detected, whereas SSTR4 (**g**) showed only weak expression. Scale bar 50 µm, images captured with CaseViewer software 40× objective.

**Figure 8 cancers-14-00162-f008:**
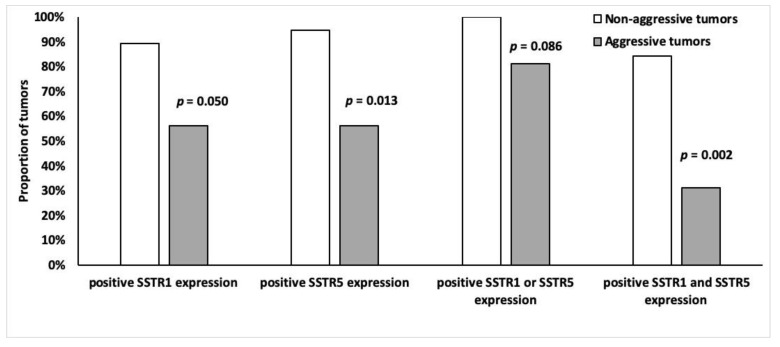
Frequency of SSTR1 and SSTR5 (%) positivity (overall score 2–4) among indolent (*n* = 19) and malignant (*n* = 16) tumors.

**Table 1 cancers-14-00162-t001:** Patient Characteristics.

No. Patients	21
No. tumors	35
Sex, Male, *n* (%)	13 (62)
Age y, mean (SD)	54.9 (18.1)
BMI, mean (SD)	25.8 (4.2)
Asymptomatic, *n* (%)	18 (86)
MEN1 syndrome, *n* (%)	7 (33)
P/S-CgA, *n* (%)	
Strongly positive	3 (14)
Weakly positive	11 (53)
Negative	7 (33)
S-PP (pmol/L), median (IQR)	91 (28.5–252.0)
S-5HIAA (nmol/L), median (IQR)	70 (51.5–95.5)
Tumor size (mm), median (IQR)	20 (10–32.5)
Tumor localization, *n* (%)	
Caput	6 (28)
Corpus	1 (5)
Cauda	10 (48)
Multiple	4 (19)
Type of surgery, *n* (%)	
Total pancreatectomy	2 (10)
Pancreaticoduodenectomy	4 (20)
Distal pancreatectomy ± splenectomy	13 (65)
Enucleation	1 (5)
Type of surgery, *n* (%)	
Open	13 (65)
Minimally invasive	7 (35)
Grade, *n* (%)	
G1	22 (63)
G2	12 (34)
G3 NEN	1 (3)

Abbreviations: P/S-CgA, plasma/serum-circulating chromogranin A; strongly positive indicates S-CgA = 13.5 nmol/L or P-CgA 9 or 37 nmol/L; weakly positive indicates S-CgA 2.2–4.7 nmol/L or P-CgA 3.0–4.8 nmol/L; negative indicates S-CgA < 2.1 nmol/L or P-CgA < 3.0 nmol/L; BMI, body mass index, kg/m^2^; MEN1, multiple endocrine neoplasia type 1 syndrome; S-PP, serum pancreatic polypeptide; S-5HIAA, serum 5-hydroxyindoleatic acid.

**Table 2 cancers-14-00162-t002:** Division of tumors into two groups according to aggressiveness.

Non-Aggressive Tumors	Aggressive Tumors
G1 tumors without any metastases	G2 tumors
	G3 tumors
	Any tumor with lymph node metastases
	Any tumor with distant metastases

**Table 3 cancers-14-00162-t003:** Somatostatin receptor (SSTR) and Ki-67 antibodies and staining protocols used for immunohistochemistry.

Antibody	Clone	Supplier	Dilution	Incubation (min)	Pre-Treatment	Detection
Ki-67	MIB-1	Dako (M7240)	1:100	32	CC1 std	ultraView
SSTR1	UMB7	Abcam ^a^ (ab137083)	1:500	45	Tris-EDTA pH 9.0	EnVision
SSTR2 ^b^	UMB1	Abcam (ab134152)	1:300	32	CC1 std	Optiview
SSTR3	UMB5	Abcam (ab137026)	1:7000	60	Citrate pH 6.0	EnVision
SSTR4	sstr4	Bio-Rad ^c^ (MCA5922)	1:500	30	Citrate pH 6.0	EnVision
SSTR5	UMB4	Abcam (ab109495)	1:1000	30	Citrate pH 6.0	EnVision

^a^ Abcam, Cambridge, UK, ^b^ This antibody was called SSTR2A in some previous studies, ^c^ Bio-Rad, Hercules, CA, USA.

**Table 4 cancers-14-00162-t004:** Summary table of somatostatin receptor (SSTR) expression association with PET/CT imaging results and clinicopathological factors.

Factor	SSTR1	SSTR2	SSTR3	SSTR4	SSTR5
[^68^Ga]Ga-DOTANOC PET/CT (positive vs. negative)	no ^a^	positivity associated, *p* = 0.043 ^b^	no ^a^	no ^a^	no ^a^
[^68^Ga]Ga-DOTANOC SUVmax	no ^a^	no ^a^	no ^a^	no ^a^	no ^a^
[^18^F]F-FDG PET/CT (positive vs. negative)	no ^a^	no ^a^	no ^a^	no ^a^	no ^a^
[^18^F]F-FDG SUVmax	no ^a^	no ^a^	no ^a^	no ^a^	membranous expression negativity associated, *p* = 0.033 ^b^
Grade	no ^a^	no ^a^	no ^a^	no ^a^	negativity associated, *p* = 0.004 ^b^
Ki-67	no ^a^	no ^a^	no ^a^	no ^a^	negativity associated, *p* = 0.038 ^b^
Lymph node SSTR status	no ^a^	no ^a^	cytoplasmic staining positively associated, *p* = 0.031 ^c^	no ^a^	no ^a^

^a^ no statistically significant association, data not shown, ^b^ data reported in the Results section, ^c^ all six primary tumors did not express cytoplasmic SSTR3 (score 0), and among lymph node metastases the median difference of the score was +2.0 (IQR 1.5–2.0).

## Data Availability

All data generated during this study are included in this article. Further enquiries can be directed to the corresponding author.

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
