# Peer review of "Correlation of Somatostatin Receptor 1–5 Expression, [^68^Ga]Ga-DOTANOC, [^18^F]F-FDG PET/CT and Clinical Outcome in a Prospective Cohort of Pancreatic Neuroendocrine Neoplasms"

_cancers, 2021, doi:10.3390/cancers14010162_

Round 1

Reviewer 1 Report

The authors address a topic that it is not novel and has been covered in the past incuding immunohistochemical expression of otehr potentially relevant receptors in a wide variety of NEN

The authors identify some limitations mainly the small number of patients and the small representation of tumors stageIV and grade 3. In addition there are several cases of MEN1 included that follow a more indolent course and have low KI67 LI. 

The correlations made are hambered by the relatively small number of cases as well as the suggestions made as the follow-up period is short

The length of the paper is excessive and both methods results and discussion could be shortened

Author Response

We thank Reviewer for all of these valuable comments. We have looked those over closely and made some significant amendments.

Reviewer comment: The authors address a topic that it is not novel and has been covered in the past incuding immunohistochemical expression of otehr potentially relevant receptors in a wide variety of NEN

The authors identify some limitations mainly the small number of patients and the small representation of tumors stageIV and grade 3. In addition there are several cases of MEN1 included that follow a more indolent course and have low KI67 LI. 

It is true that the topic is not new. However, to our knowledge, this is the first report in patients with PNEN in whom the possible correlation between SSTR PET/CT and tissue levels of all the different somatostatin receptor subtypes SSTR1–5, determined by immunohistochemistry (IHC) have been studied.  It is true that the number of G3 PNENs in this study is limited since the disease is rare. However, there was a great variety of different stages of PNETs in this cohort [six patients had stage I disease, seven had stage II disease (three IIA and four IIB), four had stage III disease and two had stage IV disease (lines 106-108, page 4 of 15)].

Our original study protocol (ClinicalTrials.gov, NCT02621541) included MEN1 patients in this study and it turned out that one third of our cohort were patients with MEN1. However, due to this valuable comment made by the reviewer we took account of this in the manuscript.

Former version: The phenomenon might be due to a high number of SSTR2 positive tumors (91%) and to the restriction to pancreatic specimens only.

Edited version: The result might be due to a high number of SSTR2 positive tumors (91%), and to a relatively high percentage of MEN1 patients having a favorable prognosis to start with. (lines 326-328, page 11 of 15).

Reviewer comment: The correlations made are hambered by the relatively small number of cases as well as the suggestions made as the follow-up period is short

Although the incidence of PNENs has increased during recent years, they are still relatively rare tumors. Considering this, we do not think that it is surprising that this cohort (n=21) is relatively small. We also think that the short follow-up time is a consequence of the prospective nature of the study. However, the rather small number of patients is acknowledged as a limitation of this study as the reviewer has noted. We have formatted this comment.

Former version: Another limitation is the relatively short follow-up time (mean 30.2 m, SD 6.2 m), which was insufficient to estimate a definitive prognosis.

Edited version: Another limitation is the relatively short follow-up time (mean 30.2 m, SD 6.2 m), which was insufficient to estimate a definitive prognosis. However, this is a consequence of the prospective nature of this study. (lines 353-355, page 12 of 15)

We have also edited the title of the study:

Former version: Correlation of somatostatin receptor 1-5 expression, 68Ga-DOTANOC, 18F-FDG PET/CT and clinical outcome in pancreatic neuroendocrine neoplasms

Edited version: Correlation of Somatostatin Receptor 1-5 Expression, [68Ga]Ga-DOTANOC, [18F]F-FDG PET/CT and Clinical Outcome in a prospective cohort of Pancreatic Neuroendocrine Neoplasms

We have now removed the comment “Our study verifies that SSTR2 is the only subtype of the five SSTRs that correlates with 68Ga-DOTANOC PET/CT.” (line 319-320, page 12, the original draft of manuscript) and added a comment “However, this must be validated in a larger cohort of PNEN patients.” (lines 347-348, page 12 of 15, the final draft of manuscript).

Reviewer comment: The length of the paper is excessive and both methods results and discussion could be shortened

The manuscript has now been shortened significantly, without impairing the contents. We have shortened Materials and Methods section 17 % and discussion 20 %. Since the journal encourages all authors of articles to share their research data, we have not removed any results. Instead we suggest that former Table 4 will be transferred to Supplementary Material. If reviewers or editor suggest some other results to be removed to supplementary material, we would certainly consider that.

We would like to point out also that this final version of the manuscript has been scrutinised and where necessary amended by the linguistic professional, Science and Medical reviser and editor Alisdair Mclean PhD.

Reviewer 2 Report

Manuscript ID: cancers-1499289

Type of manuscript: Research Article

Title: Correlation of Somatostatin Receptor 1-5 Expression, 68Ga-DOTANOC, 18F-FDG PET/CT and Clinical Outcome in Pancreatic  Neuroendocrine Neoplasms

Authors: Susanna Majala, Tiina Vesterinen, Hanna Seppänen, Harri Mustonen, Jari Sundström, Camilla Schalin-Jäntti, Risto Gullichsen, Jukka Schildt, Jukka Kemppainen, Johanna Arola  and Saila Kauhanen

Brief Summary

The main aim of the study conducted and presented by Majala and colleagues is to correlate immunohistochemical (IHC) tissue levels of five SSTR subtypes with the [68Ga]Ga-DOTANOC and [18F]F-FDG  uptake depicted by the SUVmax  in a prospective series of 21 patients (35 lesions) with PNEN. The authors describe a concordance of only SSTR subtype 2 IHC with positive/negative [68Ga]Ga-DOTANOC PET/CT findings. SSTR5 expression associates with low Ki-67. The authors objectively conclude that further studies are needed to increase the sample size and to see whether the small sample size is the reason to confirm correlation of SSTR expression profile with [68Ga]Ga-DOTANOC PET signal only for subtype 2.

General concept comments

Although the sample size is small the methodology, the study protocol and statistical tests are appropriate and well structured. Results are presented in comprehensible, logical and easily readable manner. The use of figures and tables nicely complement the text and adds to readybility of the work presented. The discussion is written in objective manner. The conclusion statements are consistent by the results presented. The references selected are appropriate and there are only few self-citations.

  Specific comments:

Lines 2, 3: 68Ga-DOTANOC, 18F-FDG should be written as [68Ga]Ga-DOTANOC, [18F]F-FDG. The authors are encouraged to harmonize the nomenclature throughout the whole manuscript with recently published paper: Consensus nomenclature rules for radiopharmaceutical chemistry — Setting the record straight (Nucl Med Biol. 2017 Dec;55:v-xi).

Line 69: “68Ga-labelled SSTR PET/” should be changed to 68Ga-labelled SST PET/CT or 68Ga-labelled SST analogues (SSAs) PET/CT

Line 317: “However, we found no correlation with uptake of 68Ga-DOTANOC.”  This is the beginning of the paragraph. It is not clear what was correlated to uptake of 68Ga-DOTANOC. Please rephrase.

Figure 1 and Figure 2 are hard to read. The letters are too small. They should be adjusted.

Author Response

We thank Reviewer for these encouraging and valuable comments. We have now formatted our manuscript according to the comments.

  1. Reviewer comment: Lines 2, 3:68Ga-DOTANOC, 18F-FDG should be written as [68Ga]Ga-DOTANOC, [18F]F-FDG. The authors are encouraged to harmonize the nomenclature throughout the whole manuscript with recently published paper: Consensus nomenclature rules for radiopharmaceutical chemistry — Setting the record straight (Nucl Med Biol. 2017 Dec;55:v-xi).

We have now formatted the manuscript to follow the instructions made by paper referred ahead and terms [68Ga]Ga-DOTANOC and [18F]F-FDG has been used.

  1. Reviewer comment: Line 69: “68Ga-labelled SSTR PET/” should be changed to 68Ga-labelled SST PET/CT or 68Ga-labelled SST analogues (SSAs) PET/CT.

We have now corrected the manuscript and this term is now presented correctly as requested “68Ga-labelled SST PET/CT” throughout the manuscript.

  1. Reviewer comment: Line 317:“However, we found no correlation with uptake of 68Ga-DOTANOC.”  This is the beginning of the paragraph. It is not clear what was correlated to uptake of 68Ga-DOTANOC. Please rephrase.

This sentence has been formatted: “However, we found no correlation between uptake of [68Ga]Ga-DOTANOC and SSTR1–5.”

  1. Reviewer comment: Figure 1and Figure 2 are hard to read. The letters are too small. They should be adjusted.

Figure 1 and 2 has been enlarged to make those more easily readable.

Due to comments given by another reviewer, the manuscript has now been condenced significantly, without impairing the contents. We have shortened Materials and Methods section 17 % and discussion 20 %. Since the journal encourages all authors of articles to share their research data, we have not removed any results. Instead we suggest that former table 4 will be transferred to supplementary material. If reviewers or editor suggest some other results to be removed to supplementary material, we would certainly consider that.

Due to reviewer comments, we have also edited the title of the study:

Former version: Correlation of somatostatin receptor 1-5 expression, 68Ga-DOTANOC, 18F-FDG PET/CT and clinical outcome in pancreatic neuroendocrine neoplasms

Edited version: Correlation of Somatostatin Receptor 1-5 Expression, [68Ga]Ga-DOTANOC, [18F]F-FDG PET/CT and Clinical Outcome in a prospective cohort of Pancreatic Neuroendocrine Neoplasms

We would like to point out also that this final version of the manuscript has been scrutinised and where necessary amended by the linguistic professional, Science and Medical reviser and editor Alisdair Mclean PhD.

Round 2

Reviewer 1 Report

The authors have addressed some of the points made. The manuscript still includes a small number of papers (MEN1 is still a confounding factor) of not all grade representation, making correlations difficult